# Adverse Childhood Experiences: Mental Health Consequences and Risk Behaviors in Women and Men in Chile

**DOI:** 10.3390/children9121841

**Published:** 2022-11-28

**Authors:** Sofía Ramírez Labbé, María Pía Santelices, James Hamilton, Carolina Velasco

**Affiliations:** 1Escuela de Psicología, Pontificia Universidad Católica de Chile, Vicuña Mackenna 4860, Macul, Santiago, 7810000, Chile; 2Centro de Investigación del Abuso y la Adversidad Temprana, CUIDA, Santiago 7810000, Chile; 3Departamento de Salud Pública, Escuela de Medicina, Pontificia Universidad Católica de Chile, Santiago 7810000, Chile; 4Escuela de Trabajo Social, Pontificia Universidad Católica de Chile, Santiago 7810000, Chile

**Keywords:** adverse experiences, childhood, mental health, risk behaviors, adulthood

## Abstract

Studies conducted worldwide indicate that adverse childhood experiences (ACEs) are among the most intense and frequent sources of stress, considerably influencing mental and physical health while also resulting in risk behaviors in adulthood. Methodology: We used data from the Pilot National Survey of Adversity and Sexual Abuse in Childhood (2020), conducted by CUIDA UC, which comprises the Adverse Childhood Experiences International Questionnaire [ACE-IQ] (Adapted). The cross-sectional methodology used made it possible to directly calculate the prevalence of adverse childhood experiences in the population sampled, at a single point in time. We performed a bivariate and univariate descriptive analysis, a correlation analysis, and a multivariate analysis, all of which will be detailed in the section entitled “General Data Analysis Procedure”. Results: We found equally high rates of adverse childhood experiences in men and women, with community violence exhibiting the highest prevalence. We found significant low- to moderate-sized associations between the multiple types of ACEs considered and mental health problems, substance use problems, criminal behaviors, and intrafamily violence (IFV), which differed between men and women. Significant correlations were detected between the ACE score and mental health, substance use, criminal behaviors, and IFV in both men and women. Importantly, ACEs were found to be predictors of all of these variables, with differences observed between men and women. Conclusions: Nearly all participants reported having had at least one ACE and more than half reported had four or more ACEs. Those who had had four or more ACEs were more likely to report problems throughout their life. Having an ACE of any type was found to be a better predictor of mental health problems and IFV in men than in women and might be a stronger risk factor for substance use and criminal behaviors in women than in men.

## 1. Introduction

Mental health problems are the main source of burden of disease for adults in Chile (20–44 years of age) [1]. This means that, in our country, mental disorders or neuropsychiatric conditions (uni and bipolar depressive disorders, schizophrenia, alcohol/drug dependency and use, anxiety disorders, epilepsy, dementia, Parkinson’s disease, migraines, cerebral palsy in children, and attention-deficit/hyperactivity disorder) constitute the leading cause of years of healthy life lost due to disability (YLD), which is associated in the long term with a major decrease in the quality of life, physical health problems, and high rates of premature mortality [2,3]. In adults, the burdens associated with neuropsychiatric conditions are mainly influenced by mood disorders and alcohol dependency. In women, unipolar depressive disorders, anxiety disorders, and assault are among the main five causes of YLDs; for men, the main causes are problems related to alcohol use and unipolar depressive disorders [2]. That is, the risk factors that impact health, and greatly decrease people’s well-being.

The latest Longitudinal Social Protection Survey conducted in Chile [4], which employed the “Patient Health Questionnaire” (PHQ-9), revealed that depression rose in the general population from 6.9% in 2015 to 16.32% between December 2019 and March 2020. These country-level findings, which demonstrate how mental health problems and risk behaviors impact people’s quality of life, stress the importance of studying and exploring the implications of certain factors that might affect mental health and behavior in adulthood.

Adverse childhood experiences (ACEs) are among several risk factors with major mental health consequences in adulthood. They have been defined as factors to which children are frequently exposed in childhood and which can interrupt development and have a negative impact on health in later years [5].

ACEs are factors with a considerable influence on both the mental and physical health in adulthood [6,7] and have been extensively studied in North America and Europe. They are internationally regarded as the most intense and frequent stress sources that children experience early in life [5,8]. A person’s history of ACEs increases their risk of suffering from emotional and behavioral problems as an adult, with its effects being deemed as adverse outcomes or consequences throughout their life [9,10]. These consequences can be varied, affecting physical health, mental health, and risk behaviors [11] in both adolescence and adulthood [12].

Even though some studies have found sex-related differences in the number of reported ACEs [13,14,15] and their mental health consequences [16,17], more evidence is needed to assert that the mental health effects of each type of adversity are not the same for men and women [18,19].

The current research on ACEs in Latin America is scarce. As of this writing, there is one epidemiological study on ACEs, family strengths, and their relationship with health-related risk factors in Argentina [20], as well as ongoing research in Chile aimed at validating and adapting the questionnaire to the sociocultural context of the country. In Chile, we have very few investigations regarding the prevalence of ACEs and their impact on the population. Therefore, and considering the impact of the consequences of adverse childhood experiences during a person’s life, it is relevant to study the association between ACEs and their effects on mental health and risk behaviors in our country while also exploring any possible sex-related differences regarding the types of adversities affecting women and men and their consequences on mental health and risk behaviors in adulthood. For this reason, this study will only include a Chilean sample.

This study was conducted as part of the 2020 Pilot National Survey of Adversity and Sexual Abuse in Childhood, carried out by CUIDA UC (Center for research on abuse and early adversity). We sought to determine the prevalence of ACEs in a sample of Chilean adults, residing in the 5th Region of Chile. Furthermore, we explored the association between ACEs and self-reported mental health throughout the life cycle while also examining the relationship between ACEs and various health-related risky behaviors, analyzing the differences between men and women. Risk behaviors will be defined as voluntary or involuntary actions performed by an individual or community which can have negative consequences, and which are also multiple and may be bio-psycho-social in nature, such as substance abuse and behaviors ultimately leading to injuries and violence [21]. These will be taken to include alcohol and substance use, criminal behaviors, and intrafamily violence.

### ACEs and the Differences between Men and Women

A meta-analysis conducted in 2018 [18] showed that there are no significant differences in sex in the association between abuse in childhood and mental health consequences in adulthood. Yet, it is worth pointing out that this study only included five articles that examined the physical and sexual abuse as causes of depressive and anxious disorders, leaving out other adverse experiences, since only these types of abuse coincided with the comparative variables of maltreatment/abuse and sex. Therefore, this evidence is insufficient for answering and solving the question of whether significant sex-related differences exist regarding physical and sexual abuse experiences or other types of early adverse experiences [19], and to establish whether significant differences exist in the association between ACEs and mental health problems in adulthood.

In 2020, a study on the role of sex in the relationship between ACEs and substance use and criminal behaviors in adolescents found no differences by sex in either the overall ACE scores or ACE accumulation, with both men and women experiencing at least one adverse experience. As for adverse experience types, no significant differences were found between men and women, except for physical abuse, which was –on average– more commonly reported by men. The correlation between ACEs and substance use was significant only for women, whereas the correlation between ACEs and criminal behaviors was significant for men [21]. Using a Brazilian cohort (*n* = 4230), researchers found that the occurrence of ACEs is positively associated with the use of alcohol, tobacco, and illicit drugs (e.g., marijuana, cocaine) in adolescents and that there are differences between men and women, with young women who have had three or more ACEs exhibiting a greater prevalence of tobacco and illicit drug use than their male peers, but with alcohol use not differing significantly between the sexes [22]. However, when studying the association between childhood trauma and alcohol dependency in adulthood (*n* = 280 patients and 137 control subjects), no significant differences were found between men and women, suggesting that emotional abuse experiences predict dependent alcohol use in adults of both sexes but without helping to establish the effect of sex and the emergence of significant differences in larger samples [23]. Using an adult sample (*n* = 34,653), authors examined the association between physical, emotional, and sexual abuse, physical and emotional neglect, and substance use (alcohol, sedatives, opiates, amphetamines, cocaine, amphetamine, cannabis, heroin, hallucinogens, and nicotine), finding that these five types of adverse experiences were associated with an increase in the problematic use in both men and women, except for physical neglect and heroin use in men and emotional neglect and amphetamine and cocaine use in men [22]. These findings are in line with previous studies that highlight the effect of abuse experiences in childhood, which have been shown to be significantly associated with the use of psychoactive drugs in adult women and men [24]. In this line of research, the authors studied the link between ACEs and alcohol use in a sample of 7279 subjects, also analyzing the moderating role of psychological stress in men and women [25]. This study found a high prevalence of substance use in both men and women who had experienced any of the ACEs considered as well as significant differences between the sexes in the association between alcohol consumption and the type of adverse experience that had affected the subjects. The authors stress that, in men, physical and sexual abuse, emotional neglect, and household dysfunctions (drug abuse, mental disease, and incarceration of a member) are linked to self-reported alcohol problems, whereas in women, emotional, physical, and sexual abuse, neglect (emotional and physical), and household dysfunctions (parental separation, drug use, and mental disease of a member) are related to alcohol problems. In addition, women were more likely to report experiences of emotional and sexual abuse, emotional neglect, parental divorce or separation, witnessing domestic violence, and having a family member with a mental disease or incarcerated, whereas men were more likely to report experiences of physical abuse and neglect [26].

Regarding the differences between men and women in the association between ACEs and aggressive behaviors, research indicates that men with more experiences of sexual abuse in childhood exhibit a stronger link to the perpetration of physical intimate partner violence as well as more antisocial behaviors than women [27]. Antisocial behaviors were also found to have a moderating effect between the experience of sexual abuse and the perpetration of violence, with the effect being greater in women than men, which suggests that the association between early sexual abuse in men and violence exerted against their partner can be mediated by other factors, such as anger, PTSD symptoms, and substance use [27]. The authors of a recent study also stressed the importance of considering both the advantages and the limitations of ACEs as an instrument, noting that it is inaccurate to state that all adverse experiences have the same influence and that it is necessary to learn more about the adverse effects of each individual experience [23].

Considering the latter, it is worth noting that Chile’s alcohol and substance use rates are high, with one in 10 people having alcohol use problems; furthermore, household drug use and marijuana dependency symptoms have increased significantly, rising from 14.8% in 2016 to 20.3% in 2018 [28]. Concerning poly-victimization in adulthood, intimate partner violence in its multiple manifestations (psychological, emotional, physical, economic, and sexual) has gained increased recognition as a global problem [29]. In Chile, a regional study examined the prevalence of this phenomenon, revealing that emotional, physical, and sexual violence are significantly greater in women [30]; also, the 4th National Survey of Violence in Family Settings (2020) found that 41.4% of women between 15 and 65 years of age have experienced IFV (physical, psychological, or sexual) at least once in their lifetime, with this rate being significantly higher than that found in 2017 [31].

Current research [19,27] has stressed the need to examine the differences between men and women, regarding the types of adverse experiences affecting them, taking into account their potential impact on the development of mental health problems and risk behaviors in adulthood, which include problematic substance use, violent behaviors toward others, and other criminal behaviors; furthermore, it is necessary to track how this relationship changes among samples and countries. Considering this, the present study examines the risk factors whose incidence and prevalence are high in Chile, according to the literature, such as alcohol and substance use, intrafamily violence, and criminal behaviors.

It is relevant to expand this line of research and apply it to the Chilean context, bearing in mind the importance of addressing early risk factors and preventing exposure to adverse experiences in childhood as a way of reducing their impact on the development of psychopathology in adulthood.

No significant differences are expected to be found in the prevalence of ACEs between men and women, but rather differences in terms of the type of experiences in childhood. Regarding sex, differences could be found in terms of mental health problems, use of OH and/or substances, criminal behavior, and IFV (greater mental health problems and complaints filed by IFV in women and greater criminal behavior, consumption of OH and /substances and complaints received by IFV by men). Finally, it is expected that the greater the number of ACEs, the greater the self-reported mental health problems will be observed throughout life, a greater number of OH and/or substance use problems, more criminal behavior, and more domestic violence will be observed.

## 2. Methodology

### 2.1. Participants and Instruments

For this research, we used the database from the Pilot National Survey of Adversity and Sexual Abuse in Childhood (2020), conducted by CUIDA UC and approved by the university’s institutional review board. The Pilot Survey was administered to an urban sample (*n* = 200) composed of 137 women (68%) and 63 men (32%) between 18 and 89 years of age who resided in the 5th Region of Chile (Concón, Quilpué, Quintero, and Villa Alemana).

*Pilot National Survey of Adversity and Sexual Abuse in Childhood:* The instrument comprises several modules. In this study, we employed the following: the demographic questionnaire, the adverse childhood experiences module, the sexual abuse module, and the late adversities module. The latter includes items on mental health, risk behaviors (criminal behaviors and alcohol and/or substance use), and family life.

*The Adverse Childhood Experiences International Questionnaire [ACE-IQ] (Adapted): Retrospective* questions focused on situations experienced before 18 years of age. The material used is based on the Adverse Childhood Experiences International Questionnaire (ACE-IQ) (The ACE-IQ was constructed by Felitti et al. (1998), who modified the conflicts tactic scale and the National Health Interview Survey (NHIS). Its internal consistency is Cronbach’s α = 0.95 [32], produced by the World Health Organization [5]. The instrument covers 13 categories in total, considering the 10 categories of the original questionnaire (physical, psychological, and sexual abuse, psychological and physical neglect, violence against a family member, mental disease affecting family members, substance abuse, incarceration of a family member, and parental separation, loss, or divorce) [6,33], plus two categories incorporated following the current research (school bullying and community violence) [34,35] and one new category of collective violence [5], which refers to contexts of political violence and armed conflicts. It is worth pointing out that the questions about sexual abuse were selected from the sexual abuse module and incorporated into the ACE-IQ. Thus, the questionnaire has 25 items plus two items belonging to the sexual abuse category.

*Self-administered Sexual Abuse Questionnaire:* Self-administered instrument, based on the NIS-3 [36,37] (Third National Incidence Study of Child Abuse and Neglect.), which classifies abuse into three types: with physical contact and penetration/rape (oral, anal, or genital penetration with the penis or anal or genital penetration with fingers or any other type of penetration), with physical contact and without penetration (acts involving any type of genital contact), and with no genital contact (caresses, exposure, or other unspecified acts with no genital contact). Based on this questionnaire, we selected items that matched the sexual abuse category of the ACE-IQ. Two questions were included: At any point in your life, has anyone groped, fondled, or kissed you through manipulation, deceit, submission, or obligation, and/or forced you to do so? Was there oral, anal, or vaginal penetration with the penis, fingers, or any other object, through manipulation, deceit, submission, or obligation and/or were you forced to do so?

*Late adversities questionnaire:* This instrument considers the modules mental health throughout the life cycle, family life, and risk behaviors (criminal behaviors and alcohol and drug use problems involving marijuana, cocaine, and cocaine paste). In the mental health throughout the life cycle section, the participants were asked to report which psychopathologies they had ever been professionally diagnosed with. The following psychopathologies were included in this section: depression, schizophrenia, post-traumatic stress, nervous anorexia or bulimia, generalized anxiety disorder, borderline personality disorder, suicidal ideation, non-suicidal self-harm, and learning disorders. Regarding the family life module, two intrafamily violence (IFV) items were considered: having lodged an intrafamily violence complaint and having been reported for such acts at least once. The questions about criminal behaviors covered the following seven problems: intentionally hurt someone in a fight, had to drop out of school or was expelled, committed theft or robbery, sexually abused or assaulted someone, trafficked or sold drugs, has been convicted by a court, has had to serve a prison sentence of more than three months. As for alcohol and/or substance use problems, the participants were asked to answer “yes” or “no” to the following statements: has had problems with alcohol, has used marijuana somewhat frequently, has used cocaine or cocaine paste, even infrequently. For this study, the following dependent variables will be considered: mental health problems, alcohol and/or substance use, criminal behaviors, and intrafamily violence.

### 2.2. Statistical Analyses

First, we constructed an ACE score for a binary version, according to the guidelines for analyzing the ACE-IQ [5]. The construction of the binary version of the ACE consists in assigning one point per category if the participant answers affirmatively (once, several times, or many times). These points are then added up, yielding a scale with scores ranging from 0 to 13. The construction of the ACE index will make it possible to conduct the necessary analyses for observing the relationships between ACE scores and their impact on mental health and health risk factors. This process was carried out using STATA 15.1. We performed descriptive bivariate and univariate analyses, correlation analyses, and multivariate analyses of the full sample, plus separate analyses for men and women. We used the chi-squared test to check for significant differences between the variables: ACEs, mental health, substance use, criminal behaviors and intrafamily violence, and sex.

To analyze the association between the variables, we used the Mathews correlation coefficient (or Phi), also considering the effect size and the significance of the relationship (*p*-value). This coefficient was used to analyze the association between each category of ACEs and each dependent variable (mental health, alcohol and/or substance use, criminal behaviors, and IFV) in the overall sample as well as in men and women, separately. To analyze the correlations between the independent variables (ACEs) and the dependent variables, we used Spearman’s correlation coefficient, given the non-normal distribution of the sample. To do so, we constructed an index for three of the dependent variables so that it would yield the number of problems of the following types reported by the participants: mental health, alcohol and/or substance use, and criminal behaviors. For the intrafamily violence (IFV) variable, we constructed a dichotomous indicator to measure the presence or absence of violence and victimization (i.e., has lodged an intrafamily violence complaint or has been legally accused of such acts). This decision was made, due to the small N of responses to these items.

Lastly, we produced a set of regression models, according to the multiple variables analyzed, considering the sex differences. We used Poisson regressions for the count variables (number of mental health problems, substance use problems, and criminal behaviors) since they are response variables with a Poisson distribution. Furthermore, we employed logistic regressions for the dichotomous variable IFV to predict the result of this categorical variable concerning the predictor variable (ACEs). It is important to emphasize that for this investigation, the 13-item ACE was used, and we incorporated the 10-item ACE to compare the prevalence with other studies that only used 10 items. 

## 3. Results

The participants were 63 men (31.5%) and 137 women (68.5%), over 18 years of age. Out of a total of 200 participants, 196 completed the adapted ACE-IQ. Table 1 shows the prevalence of ACEs considering the 13 categories used in this study [5], revealing that nearly all of the participants (88.9%) reported having had at least one adverse experience in childhood and that more than half of the participants (54.6%) reported having had four or more ACEs. Regarding the prevalence of adverse experiences by sex, no significant differences were found between men and women (*p* > 0.05), with 50.8% of the men and 56.4% of the women reporting four or more ACEs.

To compare the prevalence rates with international results, we considered the 10 categories of the original questionnaire (physical, emotional, and sexual abuse, physical and emotional neglect, parental divorce/separation, violence against a household member, family member with substance use problems, family member with severe mental health problems, incarcerated family member.) [6,33]. Table 2 shows that 80.8% of the participants reported having had at least one early adverse experience, with 76.2% of the men and 83% of the women reporting at least one type of ACE. No significant differences were found between men and women; 34.8% of the participants reported four or more ACEs, with 27% of the men and 38.5% of the women having had four or more adverse experiences in childhood. No significant differences were found between the sexes.

### 3.1. Type of Adverse Childhood Experiences by Sex

A comparison between the types of ACEs affecting men and women reveals some differences in the sample (Figure 1). Men (31.7%) report more experiences of physical neglect, community violence (63.5%), having a family member with alcohol and substance use problems (22.2%), and collective violence (33.3%), than women. For their part, women report more experiences of sexual abuse (40.1%), physical abuse (34.3%), emotional abuse (46%), emotional neglect (24.1%), bullying (47.1%), parental absence or separation/divorce (41.9%), witnessing violence in the household (48.2%), having incarcerated family members (7.3%), and having family members with mental disease (16.1%). Even though these results indicate that early adverse experiences differ by sex, the differences are only significant for sexual abuse.

Positive and statistically significant associations were found between ACEs and reporting at least one mental health problem, one substance use problem, and at least one criminal behavior in one’s lifetime. Importantly, 70.1% of the participants who reported four or more ACEs, had had at least one mental health disorder in their lifetime, 37.4% had had alcohol and/or substance use problems, and 33.6% reported having had at least one criminal behavior.

The relationship between ACEs and intrafamily violence was not found to be statistically significant. Of those who reported four or more ACEs, 18.7% had lodged an IFV complaint in their lifetime, while only 2.8% had been reported for IFV (Table 3).

### 3.2. Correlation between the Adverse Childhood Experiences and the Indexes of Mental Health, Substance Use, Criminal Behavior, and IFV

An index was created for each dependent variable that indicated the number of problems reported by the participants. The index of mental health problems indicates the number of mental health problems reported by each participant. Given the characteristics of the sample, this index ranged from 0 to six mental health problems. The index of alcohol and/or substance use problems indicates the number of problems reported by the participants, which ranged from 0 to three types of problematic use. The index of criminal behaviors indicates the number of such behaviors, ranging from 0 to eight reported behaviors. Furthermore, we considered the intrafamily violence variable as a dichotomous (yes/no) variable that indicates whether the participants had lodged an IFV report or had been reported for IFV at least once in their life.

Significant correlations were found between the number of ACEs and the number of problems reported in the full sample. A positive, moderate, and statistically significant correlation was found between the number of ACEs and the number of mental health problems. In contrast, a significant but weak correlation was found between the number of ACEs and the following consequences: alcohol and/or substance use, criminal behaviors, and IFV (Table 4).

The correlation between the number of ACEs and the number of lifetime consequences reported is significant for both men and women. For both male and female respondents, there is a positive, moderate, and significant correlation between the number of ACEs and the number of mental health, as well as a positive, weak, and significant association between the number of ACEs and the number of alcohol and/or substance use problems, criminal behaviors, and IFV events (Table 5 and Table 6).

### 3.3. Regression Models for the ACEs and the Impact on Mental Health and Health Risk Behaviors

We employed a Poisson regression model for the following count variables: mental health problems, alcohol and/or substance use, and criminal behaviors, and a logistic regression model for the dichotomous variable IFV.

The mental health problems model indicates that, for each ACE, the number of lifetime mental health problems increases by 22% in the full sample, with women exhibiting 65% more mental health problems (Table 7).

Table 8 presents the alcohol and/or substance use problems model, which indicates that, for each ACE, the number of alcohol and/or substance use problems increases by 19% in the full sample, with women exhibiting 56% less alcohol and/or substance use problems.

Regarding the criminal behaviors model, for each ACE, the lifetime number of criminal behaviors increases by 22%; in contrast, the association is negative for women, as they exhibit 64% fewer criminal behaviors (Table 9).

Lastly, the IFV model indicates that, for each ACE, the likelihood of having lodged an IFV report or having been reported for IFV increases by 34% in the full sample (Table 10).

### 3.4. Regression Model for Men and Women

A Poisson regression model was used for the count variables and a logistic regression model for the dichotomous variables to analyze the effect of ACEs on mental health problems, alcohol and/or substance use, criminal behaviors, and IFV, in both men and women.

The model of mental health problems by sex indicates that, for each ACE, the lifetime number of mental health problems in men increases by 27% (Table 11). In women, for each ACE, the lifetime number of mental health problems increases by 22%. This means that early adverse experiences are a better predictor of mental health consequences in men than in women.

The model of alcohol and/or substance use problems by sex indicates that, for each ACE, the number of such problems increases by 15% in men (Table 12), with the increase for women reaching 21%.

The model of criminal behaviors by sex indicates that, for each ACE, the lifetime number of criminal behaviors increases by 18% in men (Table 13). In contrast, for each ACE, the number of criminal behaviors increases by 24% in women. This means that ACEs are a better predictor of criminal behaviors in women than in men.

Figure 2 shows the Poisson regression models for men and women, revealing that consequences in terms of mental health, substance use, and criminal behaviors, depend on ACEs.

Lastly, the intrafamily violence model indicates that, as ACEs increase, the likelihood of having lodged an IFV report or having been reported for IFV, increases by 59% in men. In contrast, an increase in ACEs for women only increases by 28% the lifetime probability of having lodged an IFV report or having been reported for IFV (Table 14).

Figure 3 shows the logistic regression models for intrafamily violence in men and women.

## 4. Discussion

Both men and women can be affected by ACEs, with no significant differences found between them. Furthermore, the prevalence rate is high, with most of the sample (54.6%) reporting four or more ACEs. The most reported issues are community violence (58.9%), witnessing violence against a member of the household (46.5%), and being bullied (43.7%). The high prevalence of community violence experienced in childhood matches the results of the first National Survey of Poly-Victimization in Chilean Children and Adolescents (2017), which indicates that the most prevalent problem is exposure to community violence. This prompts the need to identify specific characteristics of the sociocultural context where children are growing up, at a regional and national level, given their exposure to high rates of violence from an early age.

It is relevant to note that this study analyzed the prevalence of the 10 categories of the original questionnaire [6,33] and the prevalence of 13 categories [5], incorporating two of the adversities considered by Finkelhor et al. (community violence and bullying) [34,35] plus a new category belonging to collective violence (having experienced wars, political or ethnic conflicts, or torture). A high prevalence of ACEs is observed in both cases, which is consistent with the research conducted to date. Studies indicate that most interviewees report at least one type of ACE [6,20,38], with slightly over half of the respondents reporting four or more ACEs [39]. In the present study, nearly all participants reported having had at least on adverse childhood experience, considering both the original questionnaire with 10 ACEs and the adapted instrument with 13 ACEs. However, when analyzing the prevalence of four or more ACEs considering the 10 categories, the rate decreases. Specifically, with 10 categories, 34.8% of the participants report four or more adverse childhood experiences, but when we consider all 13 categories, more than half of the participants (54.6%) report four or more ACEs, which can be ascribed to the high prevalence of community violence affecting the participants from an early age, a current issue in the Chilean society.

When considering the 13 ACE categories, the positive and significant correlations were found between adverse childhood experiences and the indexes of mental health problems, alcohol/substance use, criminal behaviors, and IFV, both in the full sample and in men and women, separately. Importantly, the strongest correlation exists between ACEs and mental health problems. It is of moderate size, being the strongest one for both the full sample and for men and women, separately. It can be observed that the participants with four or more ACEs tend to encounter more problems in their lifetime than those who report fewer ACEs. Among the participants who reported four or more ACEs, 70.1% have had at least one mental health problem in their life, that is, at least one mental health disorder diagnosed by a professional (these included: depression, generalized anxiety disorder, post-traumatic stress disorder, anorexia, bulimia, personality disorder, learning disorder, suicidal ideation, and non-suicidal self-harm). This is an alarming figure that stresses the association between ACEs and the presence of mental health problems in the Chilean population. Chile also exhibits high depression rates, above the world average; even more so, reports of mental health problems have increased during the current pandemic [40].

The findings presented in this article are partly consistent with other studies. For instance, in line with prior research [13,14,15,18,19], we found differences in the frequency of ACEs, with women reporting significantly more sexual abuse experiences. Even though men tend to report more physical neglect, community violence, collective violence, and having had a family member with alcohol and/or substance use problems at home, these differences are not significant. This stands in contrast with other studies that have indeed found significant differences between men and women [17,41]. Our findings are relevant, since the literature indicates that women report two or even three times as many adverse experiences as men [20]. We did not observe this, as both men and women reported similar ACE frequencies across all categories except for sexual abuse, which was more commonly reported by women. These findings can lay the groundwork for a discussion on the gender stereotypes that highlight women’s vulnerability to adverse and traumatic experiences, suggesting that, beyond sex and gender, childhood is a stage when both boys and girls are at risk of experiencing a similar degree of violence. Therefore, it is important to determine which cultural, social, familial, and upbringing factors are impacting the occurrence of adversities and the violation of children’s rights.

Considering that, to date, research has not yielded conclusive findings regarding the effect of ACEs and the differences between men and women, we analyzed the effect of adverse childhood experiences on mental health, substance use, criminal behaviors, and intrafamily violence and victimization, both in the full sample and in men and women, separately. This analysis revealed that ACEs are a significant predictor of mental health problems, substance use, criminal behaviors, and IFV in both men and women. These findings are supported by the literature and prior research, which indicate that experiences of physical, emotional, and sexual abuse, as well as neglect and household dysfunctions, are strong predictors of the development of mental health disorders and risk behaviors [11,42]. With respect to substance use problems, as prior research shows, early adversities are associated with a greater vulnerability to emotional dysregulation and problematic use, especially when children experience abuse, household dysfunctions, and violence [22,24,43]. This is consistent with the significant associations found in the full sample, regarding experiences of sexual abuse, living with someone with severe mental health problems, bullying, and collective and community violence. Interestingly, and consistent with the literature [9,22,44], violence and substance abuse in the household is associated with alcohol and/or substance use problems in men, for whom the present study also revealed a significant association between such problems and having lived with someone with mental health problems. In contrast, and in line with the literature, sexual abuse and parental absence or separation are associated with substance use problems in women [44], for whom this study also found a significant link between being bullied and substance use problems.

Regarding criminal behaviors, the above is in line with prior research, which indicates that people who have committed criminal acts exhibit a greater prevalence of ACEs [45], with the latter two experiences having significant associations with criminal behaviors in the present study. Concerning the impact on intrafamily violence, it is worth noting that the literature indicates that physical, sexual, and emotional abuse have a significant predictive effect regarding the perpetration of intimate partner violence and victimization [27,46]. In line with this, our study also revealed a predictive effect when considering all ACEs.

A relevant finding with respect to the impact of ACEs is that they are better predictors of mental health problems and intrafamily violence in men than in women, while also being better predictors of substance use problems and criminal behaviors in women than in men. This suggests that other intervening and moderating factors may influence and predict consequences and risk behaviors affecting health in addition to adverse childhood experiences. Certainly, these factors should be examined in future studies.

Lastly, it is necessary to highlight that being a woman and having had an ACE can produce a mental health impact comparable to being a man and having had an ACE; even more so, the effect may even be weaker. This is relevant for clinical practice and at a theoretical-comprehensive level, regarding the historical biases associated with gender, which have led to the mistaken belief that being a woman is a risk factor for suffering adverse experiences and being affected by negative health consequences. The evidence produced by this study can be greatly relevant for illustrating the scope of the effects of adverse experiences on mental health and risk behaviors in a population of adults in Chile. Furthermore, our findings stress the need to analyze which other intervening factors could explain these types of health consequences, both in men and women, while also considering people’s subjective experience and the singular effect that an ACE can have on each individual. We expect that these findings will improve our knowledge about the prevalence of early adverse experiences and their effect on people’s quality of life in Chile, thus enriching public policy by informing prevention and reparation strategies.

## Figures and Tables

**Figure 1 children-09-01841-f001:**
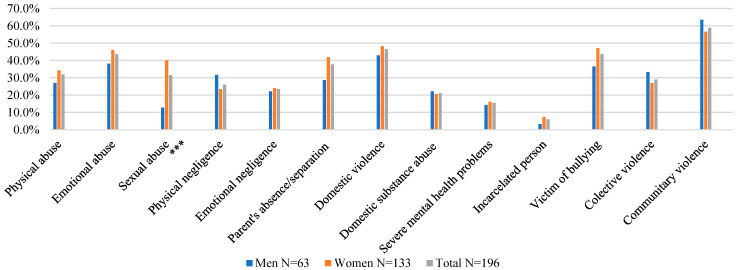
Adverse Childhood Experiences in Men and Women. *p* > 0.0001 ***.

**Figure 2 children-09-01841-f002:**
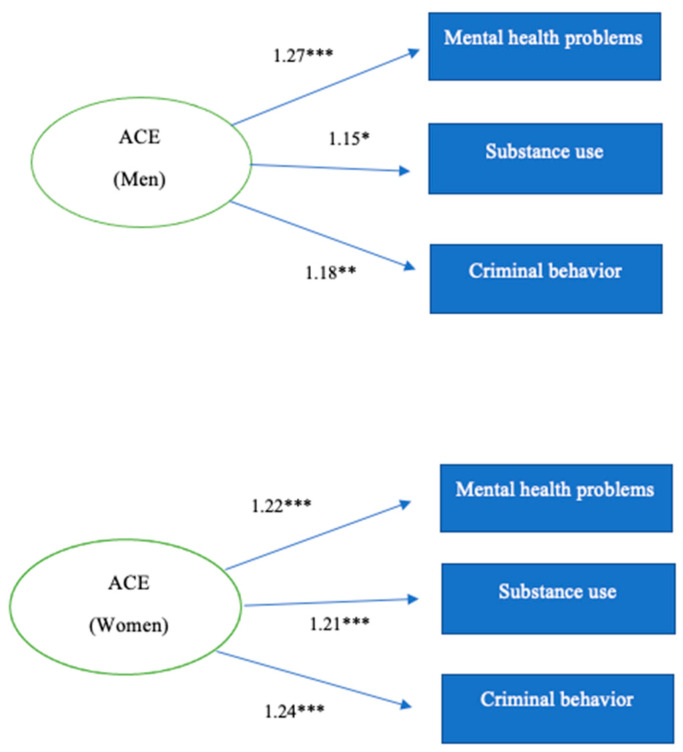
Regression models: impact on mental health, alcohol and/or substance use, and criminal behaviors in men and women. *p* < 0.0001 *** *p* < 0.001 *** p <* 0.01 *.

**Figure 3 children-09-01841-f003:**
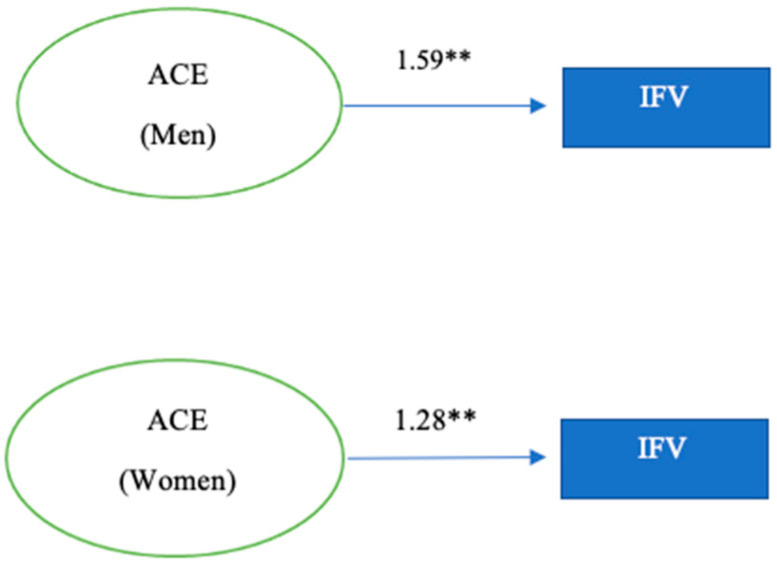
Regression model: intrafamily violence in men and women *p* < 0.001 ****.

**Table 1 children-09-01841-t001:** Prevalence of ACEs.

	Men N = 63 %	Women N = 133 %	Total N = 196 %
ACE			
0	9.50%	10.50%	10.20%
1	12.70%	10.50%	11.20%
2	17.50%	10.50%	12.80%
3	9.50%	12.00%	11.20%
1 or more	90.5%	89.5%	88.9%
4 or more	50.80%	56.40%	54.60%

*Note*: The table shows the prevalence of adverse childhood experiences by sex in the full sample, considering the 13 categories. Differences between men and women are not significant.

**Table 2 children-09-01841-t002:** Prevalence of ACEs considering the 10 categories.

	Men N = 63 %	Women N = 135 %	Total N = 198 %
ACE			
0	23.8%	17.0%	19.2%
1	14.3%	20.0%	18.2%
2	20.6%	10.4%	13.6%
3	14.3%	14.1%	14.1%
1 or more	76.2%	83.0%	80.8%
4 or more	27.0%	38.5%	34.8%

*Note*: The table shows the prevalence of adverse childhood experiences by sex in the full sample, considering the 10 categories. Differences between men and women are not significant.

**Table 3 children-09-01841-t003:** Adverse Childhood Experiences and at least one Mental Health Problem, Substance Use Problem, Criminal Behavior, and IFV.

			ACE			
	0	1	2	3	4 or More	Total ACE
Mental Health Problems						
Yes	15.00%	18.20%	20.00%	36.40%	**70.1%**	48.50%
No	85.00%	81.80%	80.00%	63.60%	29.90%	51.50%
Substance use Problem						
Yes	5.00%	13.60%	20.00%	27.30%	**37.40%**	28.10%
No	95.00%	86.40%	80.00%	72.70%	62.60%	71.90%
Criminal Behavior						
Yes	0.00%	9.10%	12.00%	36.40%	**33.60%**	25.00%
No	100.00%	90.90%	88.00%	63.60%	66.40%	75.00%
Filing IFV Complaint						
Yes	0.00%	4.50%	12.00%	13.60%	**18.70%**	13.80%
No	100.00%	95.50%	88.00%	86.40%	81.30%	86.20%
Receiving IVF Complaint						
Yes	0.00%	0.00%	0.00%	0.00%	**2.80%**	1.50%
No	100.00%	100.00%	100.00%	100.00%	96.30%	98.00%

*Note*: The figure shows the percentage of participants who reported 4 or more ACEs and at least one mental health problem, at least one substance use problem, at least one criminal behavior, and IFV in their lifetime.

**Table 4 children-09-01841-t004:** Correlation between ACEs and consequences for mental health, criminal behaviors, alcohol and/or substance use, and IFV in the full sample.

	ACE	Mental Health	Substance Use	Criminal Behaviors	IVF
ACE	1.0				
Mental health	0.55 ***	1.0000			
Substance use	0.2692 ***	0.2052 **	1.0000		
Criminal behaviors IVF	0.2929 *** 0.2930 ***	0.2576 ** 0.1621 *	0.4309 *** 0.0130	1.0000 0.0725	1.0000

*Note*: The table indicates Spearman’s correlation coefficients and significance. *p* > 0.0001 *** *p* < 0.001 ** *p* < 0.01 *.

**Table 5 children-09-01841-t005:** Correlation between ACEs and impact on mental health, criminal behaviors, alcohol and/or substance use, and IFV in men.

	ACE	Mental Health	Substance Use	Criminal Behaviors	IVF
ACE	1.0000				
Mental health	0.4566 ***	1.0000			
Substance use	0.3073 *	0.2156	1.0000		
Criminal behaviors IVF	0.3706 ** 0.3833 **	0.2829 * 0.2137	0.4089 *** 0.0424	1.0000 0.2556	1.0000

*Note*: The table indicates Spearman’s correlation coefficients and significance. *p* > 0.0001 *** *p* < 0.001 ** *p* < 0.01 *.

**Table 6 children-09-01841-t006:** Correlation between ACEs and impact on mental health, criminal behaviors, alcohol and/or substance use, and IFV in women.

	ACE	Mental Health	Substance Use	Criminal Behaviors	IVF
ACE	1.0000				
Mental health	0.5816 ***	1.0000			
Substance use	0.3119 ***	0.2656 **	1.0000		
Criminal behaviors IVF	0.3142 *** 0.2607 **	0.3352 *** 0.1334	0.4311 *** 0.0128	1.0000 0.0008	1.0000

*Note*: The table indicates Spearman’s correlation coefficients and significance. *p* > 0.0001 *** *p* < 0.001 **.

**Table 7 children-09-01841-t007:** Poisson model for the mental health problems.

	b	SE	Exp (b)
Intercept	−1.36 ***	0.2	0.26
ACE	0.20 ***	0.02	1.22
Sex (women)	0.50 ***	0.19	1.65

*Note*: *p* > 0.0001 ***.

**Table 8 children-09-01841-t008:** Poisson Model for the alcohol and/or substance use problems.

	b	SE	Exp (b)
Intercept	1.34 ***	0.25	0.26
ACE	0.17 ***	0.04	0.19
Sex (women)	−0.82 **	0.24	0.44

*Note*: *p* > 0.0001 *** *p* < 0.001 **.

**Table 9 children-09-01841-t009:** Poisson model for criminal behaviors.

	b	SE	Exp (b)
Intercept	−1.39 ***	0.26	0.25
ACE	0.2 ***	0.04	1.22
Sex (women)	−1.01 **	0.24	0.36

*Note*: *p* > 0.0001 *** *p* < 0.001 **.

**Table 10 children-09-01841-t010:** Logistic regression model for intrafamily violence.

	b	SE	Odds Ratio
Intercept	−3.27 ***	0.54	0.04
ACE	0.29 ***	0.07	1.34
Sex (women)	0.09 **	0.47	1.1

*Note*: *p* > 0.0001 *** *p* < 0.001 **.

**Table 11 children-09-01841-t011:** Poisson model for mental health problems by sex.

	b	SE	Exp (b)
Intercept	−1.57 ***	0.36	0.2
ACE Men	0.24 ***	0.06	1.27
Intercept	−0.82 ***	0.17	0.44
ACE Women	0.2 ***	0,02	1.22

*Note*: *p* > 0.0001 ***.

**Table 12 children-09-01841-t012:** Poisson model for alcohol and/or substance use problems by sex.

	b	SE	Exp (b)
Intercept	−1.19 ***	0.33	0.3
ACE Men	0.14 *	0.06	1.15
Intercept	−2.29 ***	0.35	0.1
ACE Women	0.19 ***	0.05	1.21

*Note*: *p* > 0.0001 *** *p* < 0.01 *.

**Table 13 children-09-01841-t013:** Poisson model for criminal behaviors by sex.

	b	SE	Exp (b)
Intercept	−1.24 ***	0.34	0.29
ACE Men	0.17 **	0.06	1.18
Intercept	−2.54 ***	0.38	0.08
ACE Women	0.22 ***	0.05	1.24

*Note*: *p* > 0.0001 *** *p* < 0.001 **.

**Table 14 children-09-01841-t014:** Logistic regression model for IFV by sex.

	b	SE	Odds Ratio
Intercept	−1.422 ***	1.03	0.01
ACE Men	0.47 **	0.16	1.59
Intercept	−2.86 ***	0.51	0.06
ACE Women	0.25 **	0.08	1.28

*Note*: *p* > 0.0001 *** *p* < 0.001 **.

## Data Availability

The data presented in this study are available upon request from the corresponding author.

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
