# Peer review of "Adverse Childhood Experiences: Mental Health Consequences and Risk Behaviors in Women and Men in Chile"

_children, 2022, doi:10.3390/children9121841_

Round 1
Reviewer 1 Report
This manuscript presents prevalence rates of ACEs in a Chilean sample, as well as associations between ACEs and outcomes in a Chilean sample. This is an important cultural context to be represented in the ACEs literature, but there are some changes needed in the manuscript presentation and data analysis.
1. Abstract: Please define IFV before abbreviating it
2. Introduction: The introduction is very long and goes into a lot of specific detail about the impact of particular adverse experiences (e.g., certain types of abuse) on mental health and other outcomes. I think the introduction can be made much more concise. For example, authors often spend multiple sentences discussing a single article, when they could discuss trends among findings from multiple articles to save space and help the reader understand the literature as a whole.
3. Introduction: At the end of the intro where the authors discuss the current study, I’d like to see some hypotheses listed. Is this study exclusively exploratory, or are there certain gender differences the authors expect based on Chilean culture and gender norms in Chile?
4. Methods – Please specify which ACEs (the 10-item or the 13-item) will be used for the regression analysis to assess the impact of ACEs on mental health, criminal behavior, etc.
5. Results – The first set of regression models done to assess associations between ACEs and outcomes include sex as a covariate, so there is no need to do a second set of regressions separating male ACEs from female ACEs. In fact, doing separate regressions for each sex prevents the direct comparison of the sexes, which is why some of the results of the split regressions contradict the results of the initial regressions that include sex as a covariate. I would keep the initial set of regressions using sex as a covariate and delete the sex-specific regressions the split male and female ACEs.
6. Results – Much of the introduction talks about the impact of specific types of adversity on outcomes, so I was surprised that the analysis in this paper focused on a single ACEs sum score. I think it would be interesting, and in keeping with the literature reviewed, to split ACEs by child maltreatment and family dysfunction (in keeping with the original 10-item ACEs) and put them in the same regression model to see if they differentially predict your outcomes. You could also look at community violence and collective violence together as a sub-type of ACEs representing community-level adversities. This would allow for a more specific discussion of the impact of different ACEs on outcomes in Chile.
7. Discussion (and Intro): The focus of this study and the most novel aspect of it is exploring associations between ACEs and outcomes in a novel cultural sample, Chilean adults. However, I feel that the paper is missing a discussion of the connection between the results and Chilean culture and/or gender norms. Why did the authors think it was necessary to explore these relationships in Chile specifically – were there certain differences they were expecting? I think more cultural connections need to be made in the introduction and discussion sections.
8. Discussion (& Results): One of the more novel aspects of this study is the inclusion of a collective violence item. I think there could be more focus on the prevalence of this item specifically, why it was added and maybe even analysis to see if collective violence accounts for outcomes over and above other kinds of ACEs more typically measured in other countries.
Author Response
- Abstract: Please define IFV before abbreviating it
Addressed.
- Introduction: The introduction is very long and goes into a lot of specific detail about the impact of particular adverse experiences (e.g., certain types of abuse) on mental health and other outcomes. I think the introduction can be made much more concise. For example, authors often spend multiple sentences discussing a single article, when they could discuss trends among findings from multiple articles to save space and help the reader understand the literature as a whole.
Addressed.
- Introduction: At the end of the intro where the authors discuss the current study, I’d like to see some hypotheses listed. Is this study exclusively exploratory, or are there certain gender differences the authors expect based on Chilean culture and gender norms in Chile?
Hypothesis were made for this study, and we have now included them in the manuscript:
- No significant differences are expected to be found in the prevalence of ACEs between men and women, but in terms of the type of experiences in childhood.
- It is expected to find differences by sex in terms of mental health problems, use of OH and/or substances, criminal behavior and IFV (Greater mental health problems and complaints filed by IFV in women and greater criminal behavior, consumption of OH and / substances and complaints received by IFV by men).
-It is expected that the greater the number of ACEs, the greater self-reported mental health problems will be observed throughout life, a greater number of OH and/or substance use problems, more criminal behavior, and more domestic violence.
- Methods – Please specify which ACEs (the 10-item or the 13-item) will be used for the regression analysis to assess the impact of ACEs on mental health, criminal behavior, etc.
Addressed
- Results – The first set of regression models done to assess associations between ACEs and outcomes include sex as a covariate, so there is no need to do a second set of regressions separating male ACEs from female ACEs. In fact, doing separate regressions for each sex prevents the direct comparison of the sexes, which is why some of the results of the split regressions contradict the results of the initial regressions that include sex as a covariate. I would keep the initial set of regressions using sex as a covariate and delete the sex-specific regressions the split male and female ACEs.
The first set of regressions cannot be eliminated since they account for the % of health problems, criminal behavior and violence that predict ACEs in the total sample, and the second set does so in terms of gender differences.
- Results – Much of the introduction talks about the impact of specific types of adversity on outcomes, so I was surprised that the analysis in this paper focused on a single ACEs sum score. I think it would be interesting, and in keeping with the literature reviewed, to split ACEs by child maltreatment and family dysfunction (in keeping with the original 10-item ACEs) and put them in the same regression model to see if they differentially predict your outcomes. You could also look at community violence and collective violence together as a sub-type of ACEs representing community-level adversities. This would allow for a more specific discussion of the impact of different ACEs on outcomes in Chile.
This point is presented in the discussion for future research since it exceeds the objectives of this present investigation.
- Discussion (and Intro): The focus of this study and the most novel aspect of it is exploring associations between ACEs and outcomes in a novel cultural sample, Chilean adults. However, I feel that the paper is missing a discussion of the connection between the results and Chilean culture and/or gender norms. Why did the authors think it was necessary to explore these relationships in Chile specifically – were there certain differences they were expecting? I think more cultural connections need to be made in the introduction and discussion sections.
Addressed. Cultural reference is made regarding the prevalence of community violence, which is a current issue in Chile; sexual abuse; and mental health impacts in the conclusion.
- Discussion (& Results): One of the more novel aspects of this study is the inclusion of a collective violence item. I think there could be more focus on the prevalence of this item specifically, why it was added and maybe even analysis to see if collective violence accounts for outcomes over and above other kinds of ACEs more typically measured in other countries.
Addresed.
Reviewer 2 Report
In the present manuscript, the authors reported on the effects of adverse childhood experiences on mental health. The topic fits the scope of the journal and has potential implications for public health. However, there are several important issues that need to be addressed.
1. The introduction should be re-structured. The "Background" section should not be a subsection; rather, it needs to be integrated into the Introduction. Moreover, the introduction section should be more focused on the main research question, and the history of the studies on ACEs is less relevant.
2. The sample was in a specific country, which may compromise the validity of the findings. So, the reason why the sample was focused and how the sample in the country is different from other countries should be explicitly stated.
3. Based on the Method section, the study did not employ secondary data. Further, the data is not national representative dataset. Please clarify these points.
4. Given no gender differences in ACE effects on mental health, why were stratified analyses used and separate males and females?
5. The results were a bit overwhelming, so the analyses and results need to trimmed to be more focused. The logistic regressions should be the core analysis.
6. The study was exploratory, and no hypotheses were made, which impact the validity of the findings.
Author Response
- The introduction should be re-structured. The "Background" section should not be a subsection; rather, it needs to be integrated into the Introduction. Moreover, the introduction section should be more focused on the main research question, and the history of the studies on ACEs is less relevant.
The introduction has been restructured and is now focused only on the main information.
- The sample was in a specific country, which may compromise the validity of the findings. So, the reason why the sample was focused and how the sample in the country is different from other countries should be explicitly stated.
This has now been explicitly stated.
- Based on the Method section, the study did not employ secondary data. Further, the data is not national representative dataset. Please clarify these points.
This has been clarified.
- Given no gender differences in ACE effects on mental health, why were stratified analyses used and separate males and females?
There are indeed gender differences, since the ACEs would be predicting a higher percentage of mental health problems in men than in women. There are no differences in the prevalence of ACEs in men and women.
- The results were a bit overwhelming, so the analyses and results need to trimmed to be more focused. The logistic regressions should be the core analysis.
The results section has been revised.
- The study was exploratory, and no hypotheses were made, which impact the validity of the findings.
Hypothesis were made for this study, and we have now included them in the manuscript:
- No significant differences are expected to be found in the prevalence of ACEs between men and women, but in terms of the type of experiences in childhood.
- It is expected to find differences by sex in terms of mental health problems, use of OH and/or substances, criminal behavior and IFV (Greater mental health problems and complaints filed by IFV in women and greater criminal behavior, consumption of OH and / substances and complaints received by IFV by men).
-It is expected that the greater the number of ACEs, the greater self-reported mental health problems will be observed throughout life, a greater number of OH and/or substance use problems, more criminal behavior, and more domestic violence.
Round 2
Reviewer 1 Report
Thank you for your continued work on this paper and your responses to the suggested edits. There are a few more specific ways the paper can be improved: 1) The regression models for men and women, I believe, are trying to answer the question: do ACEs predict X outcome (e.g., mental health) differently by sex? I think this could best be answered by adding an interaction term (ACES*sex) into your regression model, rather than doing a male regression model and a female regression model for each outcome. This would decrease the total number of regressions being completed, reducing potential Type 1 error inflation. 2) Please review the paper for syntax, grammar and other errors. There are places in the manuscript where the same figure is shown twice in a row and overall the writing could be improved.
Author Response
Dear reviewer,
Thank you for your comments. We have thoroughly reviewed the article and included your suggestions.
1) We returned to the data and discussed this point with the team. We decided to separate men and women because of the characteristics of the variables. The ACEs variable is a count variable and the sex variable is dichotomic. If we use men and women separately, it is a count variable and can be calculated with ACEs.
2) We have corrected the grammar and spelling in the article.
Reviewer 2 Report
The authors have addressed issues raised by the reviewer. The introduction has been reorganized, which is clearer and easier for readers to follow. The important theories have been covered. In the method section, the tables and descriptions have been adjusted, which present the results in a precise and succinct way. Overall, the manuscript is improved and would be suggested to publish on the journal.
Author Response
Dear reviewer,
Thank you for your comments. We have thoroughly reviewed the article and checked for spelling and grammar.